# Lifting the bandwidth limit of optical homodyne measurement with broadband parametric amplification

Yaakov Shaked[1], Yoad Michael[1], Rafi Z. Vered[1], Leon Bello[1], Michael Rosenbluh[1] & Avi Pe'er [1]

Homodyne measurement is a corner-stone method of quantum optics that measures the quadratures of light—the quantum optical analog of the canonical position and momentum. Standard homodyne, however, suffers from a severe bandwidth limitation: while the bandwidth of optical states can span many THz, standard homodyne is inherently limited to the electronically accessible MHz-to-GHz range, leaving a dramatic gap between relevant optical phenomena and the measurement capability. We demonstrate a fully parallel optical homodyne measurement across an arbitrary optical bandwidth, effectively lifting this bandwidth limitation completely. Using optical parametric amplification, which amplifies one quadrature while attenuating the other, we measure quadrature squeezing of 1.7 dB simultaneously across 55 THz, using the pump as the only local oscillator. As opposed to standard homodyne, our measurement is robust to detection inefficiency, and was obtained with >50% detection loss. Broadband parametric homodyne opens a wide window for parallel processing of quantum information.

[1] Department of Physics and BINA Center of Nano-technology, Bar-Ilan University, 52900 Ramat-Gan, Israel. Correspondence and requests for materials should be addressed to A.P'e. (email: avi.peer@biu.ac.il)

The standard representation of a nearly monochromatic light field is either as a complex amplitude $a = |a|e^{i\varphi}$ to reflect the amplitude and phase of the field oscillation $E(t) = ae^{-i\Omega t} + a^* e^{i\Omega t} = |a|\cos(\Omega t + \varphi)$ ($\Omega$ the carrier frequency), or as a superposition of two quadrature oscillations $E(t) = x \cos \Omega t + y \sin \Omega t$, where $x = a + a^*$ and $y = i(a - a^*)$ are the real quadrature amplitudes of the cosine-wave and sine-wave components. While the quadrature representation may be just a mathematical convenience in classical electromagnetism, it is of fundamental importance in quantum optics. The two quadrature operators $x = a + a^\dagger$ and $y = i(a - a^\dagger)$ form a conjugate pair of non-commuting observables ($[x, y] = 2i$) analogous to position and momentum in mechanics, indicating that their fluctuations are related by quantum uncertainty $\Delta x \Delta y \geq 1$. This conjugation is most emphasized with quantum squeezed light[1], where the quantum uncertainty of one quadrature amplitude is reduced (squeezed), while the uncertainty of the other is inevitably increased (stretched), that is, $\Delta x < 1 < \Delta y$[2–4].

Homodyne measurement, which extracts the quadrature information of the field, forms the backbone of coherent detection in physics and engineering, and plays a central role in quantum information processing, from measuring non-classical squeezing[1], through quantum state tomography[5–7], generation of non-classical states[8], quantum teleportation[9–11], quantum key distribution, and quantum computing[12,13]. To measure the field quadratures, homodyne detection compares the optical signal to a strong and coherent quadrature reference (local oscillator—LO), where the specific quadrature axis to be measured is selected by tuning the phase of the LO. Hence, the heart of a homodyne detector encompasses an external LO and a field multiplier. This is most evident for homodyne measurement in the radio-frequency (RF) domain, where the input radio-wave and the LO are directly multiplied using an RF frequency mixer. In optics, however, direct frequency mixers do not exist. Instead, standard optical homodyne relies on a beam splitter to superpose the optical input and the LO (see Fig. 1a) and on the nonlinear electrical response of square-law photo-detectors as the field multipliers that generate an electronic signal proportional to the measured $x$ or $y$ quadrature. Thus, measuring quadratures with standard homodyne is inherently limited to the electronic bandwidth of the photo-detectors (MHz-to-GHz). In addition, homodyne detection is highly sensitive to the noise level and quantum efficiency of the detectors, which leads to decoherence due to the addition of vacuum noise[14–17].

Yet, optical states of light can easily span optical bandwidths of 10–100 THz and more, where the quadratures $x(t)$, $y(t)$ vary rapidly on a time scale comparable to the optical cycle ($E(t) = x(t) \cos \Omega t + y(t) \sin \Omega t$). Thus, the detection method enforces an inherent distinction between nearly monochromatic and broadband fields. In the near monochromatic case, the instantaneous quadrature amplitudes vary slowly over millions of optical cycles, and can be directly observed from the time-dependent electronic signal of the homodyne output. For broadband light, however, photo-detectors are too slow to follow the quadrature variations, demanding an inherently different measurement approach[14–17].

Two examples can illuminate both the potential utility of broad bandwidth in quantum information and the difficulty of standard methods to exploit it. One example is one-way quantum computation with a quantum frequency comb[13,18], which forms the most promising realization of scalable quantum information to date. This approach exploits the large bandwidth of frequency mode pairs from a single parametric oscillator (two-mode squeezed vacuum) as a set of quantum modes (Q-modes), where coupling among near Q-modes demonstrated the largest entangled cluster states to date along with a complete set of quantum gate operations[13]. The number of parallel Q-modes is dictated by the squeezing bandwidth of the parametric oscillator, which can extend up to a full optical octave by rather simple means (limited only by phase matching of the nonlinear interaction)[19–21]. Assuming a squeezing bandwidth of 10–100 THz, the number of simultaneous Q-modes can easily exceed $10^5$. The limitation of this approach to quantum computation is the bandwidth of the measurement, where each Q-mode requires a separate homodyne detection using a precise pair of phase-correlated LOs. A broad bandwidth of Q-modes requires a dense set of correlated LOs and multiple homodyne measurements, quickly multiplying the complexity to impracticality. In our experiment, we simultaneously measure the entire bandwidth of a broadband two-mode squeezed vacuum with only one LO—the pump field that generates the squeezed light to begin with.

Another example is in quantum key distribution, where enhanced bandwidth was employed to increase the data rate by increasing the number of bits per photon. The concept here is to divide the photon readout time, which is limited by photo-detectors, into multiple short time-bins, which act as an additional time stamp for each photon (or pair)[22,23]. The time stamp (bin), which is usually detected using a Franson interferometer[24], enhances the number of bits per photon to $\log_2 N$, where $N$ is the number of time-bins. Theoretically, if the bandwidth limit of the detector could be lifted, all time (or frequency) bins could be detected independently, and a $\times N$ higher flux of photons could be used, allowing full parallelization of the communication across the available bandwidth and enhancement of the data throughput by a larger factor $N$ (compared to $\log_2 N$).

Here we present a different approach to optical homodyne, which resorts to a broadband optical nonlinearity—parametric amplification, as the field multiplier. Using this method we measure the entire bandwidth simultaneously with a single homodyne device and a single LO. Specifically, since parametric gain only amplifies one quadrature of the input signal but attenuates the other, analysis of the output spectrum enables evaluation of the input quadratures. Due to the parametric amplification of the quadrature of interest, our measurement is insensitive to detection inefficiency (and to the added vacuum noise it introduces). Indeed, our observation of broadband squeezing was easily obtained with >50% loss in the detection channel. With sufficient parametric gain, any given $x$ quadrature can be amplified to overwhelm the attenuated orthogonal $y$ quadrature, even if it was originally squeezed, such that the resulting output signal is practically proportional only to the input $x$ quadrature. Even if the parametric gain in the measurement is not high enough to completely diminish the $y$ quadrature, measurement is simple, once the desired $x$ quadrature is sufficiently enhanced above the vacuum level. Specifically, two orthogonal measurements, one for each quadrature, enable extraction of both quadratures (average) over the entire optical bandwidth, as detailed hereon.

## Results

**Experiment**. The basic concept of our method for broadband homodyne detection is illustrated in Fig. 1, showing in Fig. 1a the standard homodyne method and in Fig. 1b the parametric homodyne detection as realized by a broadband parametric amplifier acting on the quadratures of the light. To describe the effect of the parametric amplifier in Fig. 1b, we use the common expression for the optical field at the output of a parametric amplifier (based on either three-wave or four-wave mixing (FWM) optical nonlinearity: $a_{out} = a_{in}\cosh(g) + a_{in}^\dagger \sinh(g) = x_{in}e^g + y_{in}e^{-g}$, where $a_{in}, a_{in}^\dagger$ are the input field operators, $x_{in}, y_{in}$ are the input quadratures, and $g$ is the parametric gain. Hence, the parametric amplification amplifies one input quadrature ($x_{in}e^g$)

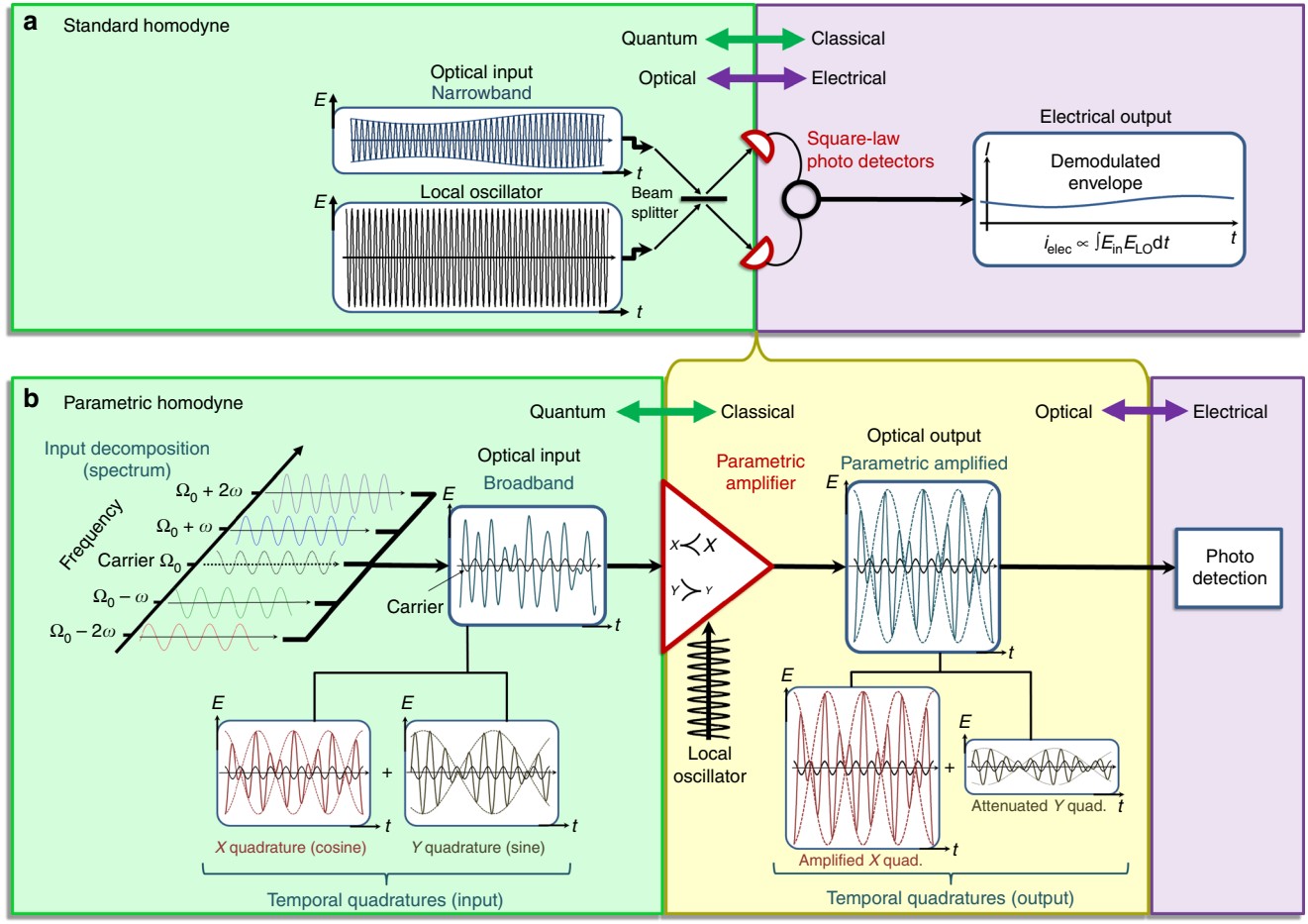

**Fig. 1** Parametric homodyne vs. standard homodyne. **a** The standard homodyne measurement, where the electrical nonlinearity of two square-law photo-detectors in a balanced detection scheme is used to multiply the quantum input field with the LO. The photo-detection (and the homodyne gain associated with it) mark the transition from a quantum optical input (green frame) to a classical and electrical output (purple frame), where a severe bandwidth limitation is imposed by the electronic detection. **b** The parametric homodyne method uses an optical parametric amplifier to amplify the quantum quadrature of interest to a classical level (and attenuate the other) already in the optical domain before the photo-detection, thereby generating an optical intermediate signal (light yellow frame), which is already classical, yet preserves the full optical bandwidth of the quantum input, and allows parallel photo-detection and classical manipulation over the entire bandwidth. In the illustration, the broadband optical input is composed of four frequency components (two-mode pairs), such that the $\Omega \pm 2\omega$ mode-pair generates a fast beat envelope only on the x quadrature, and the $\Omega \pm \omega$ mode-pair generates a slower beat envelope only on the y quadrature. The total broadband optical field conceals the quadrature information, but after the parametric amplifier, the x quadrature is amplified and the y quadrature is attenuated, such that the resulting parametric output is almost entirely proportional to the x quadrature, rendering the fast x beat clearly visible. The small black oscillation plotted on top of the various fields is a cosine local-oscillator reference to identify the x and y quadratures

while attenuating the other ($y_{in}e^{-g}$), indicating that for sufficient amplification, the output field reflects one quadrature of the input primarily without adding noise to the measured quadrature, thus offering a quadrature selective quantum measurement. The amplification process responds instantaneously to time variations of the quadrature amplitudes $x(t)$, $y(t)$ and the amplification bandwidth is limited only by the phase matching conditions in the nonlinear medium, which can easily span an optical band-width of 10–100 THz (implications of the time dependence are deferred to a later discussion). In our experiment, we measured the spectrally resolved intensity of the chosen input quadrature $x^{\dagger}(\omega)x(\omega)$ simultaneously across the entire bandwidth by detecting the output spectrum of a parametric amplifier with an input of broadband squeezed vacuum.

We note that the parametric amplifier used in the measurement need not be ideal. Specifically, since the attenuated quadrature is not measured, it is not necessarily required to be squeezed below vacuum, only to be sufficiently suppressed

compared to the amplified quadrature. Consequently, restrictions on the measurement amplifier are considerably relaxed compared to sources of squeezed light, allowing it to operate with much higher gain.

The common source for squeezed light or squeezed vacuum, in our experiment, is also a parametric amplifier. If the amplification is spontaneous (vacuum input), the amplifier attenuates one of the quadratures of the vacuum input state, squeezing its quantum uncertainty. For measuring the squeezing, we exploit the same nonlinearity and the same pump that generates the squeezed state in the first place, thus guaranteeing a bandwidth match of the homodyne measurement to the squeezing process. The quad-rature information over a broad frequency range is obtained simultaneously by measuring the spectrum of the light at the output of the detection parametric amplifier. With a single LO—the pump, each individual frequency component is measured independently, and the number of accessible Q-modes (or Q-bits) that could be utilized simultaneously would be multiplied by N

**Fig. 2** Experimental schematic of the parametric homodyne. The experiment consists of two parts: (1) generation of broadband squeezed light and (2) homodyne measurement of the generated squeezing. Broadband two-mode squeezed light is generated via spontaneous four-wave mixing (FWM) in a photonic crystal fiber (PCF) pumped by 12 ps laser pulses (786 nm). After generation, the pump is replaced by an appropriately delayed copy of the original pump light, via a narrowband filter, which allows independent intensity and phase control, to tune the parametric gain and to select the specific quadrature to be measured. Then, the new pump and the FWM light enter the second PCF for the homodyne measurement. After this second (measurement) pass through the amplifier, the pump is separated from the FWM light by a narrowband filter and the FWM light is measured by a spectrometer

(the number of resolved frequency bins) rather than $\log_2 N$. As will be explained later, a single-frequency component of the quadrature is actually a combination of two-frequency modes, commonly termed signal $\omega_s$ and idler $\omega_i$, symmetrically separated around the main carrier frequency $\Omega$.

The experimental demonstration of broadband parametric homodyne consists of two parts (see Fig. 2): first, generation of broadband squeezed vacuum, and second, parametric homodyne detection of the generated squeezing. We generate broadband squeezed vacuum by collinear FWM in a photonic crystal fiber (PCF) that is pumped by narrowband picosecond pulses near the zero dispersion wavelength of the PCF. To measure the generated squeezing, we couple the light generated by the FWM process together with the pump into another PCF, which acts as the measurement parametric amplifier (in the experiment this was the same PCF in the backward direction). After this second (measurement) pass we record the parametric output spectrum to extract the quadrature information (see Fig. 3a).

Since squeezed vacuum is a gaussian state, its quadrature distribution is completely defined by the second moment. We therefore measure the average spectral intensity (with averaging times of a few 10 ms) and reconstruct the average quadrature fluctuations $\langle x^\dagger x \rangle, \langle y^\dagger y \rangle$. Measurement of the instantaneous intensity distribution is possible with a shorter integration time, but not necessary for squeezed vacuum.

Fringes appear across the output spectrum of the measurement parametric amplifier due to chromatic dispersion in the optical components (filters, windows, etc.), which introduces a varying spectral phase with respect to the pump across the FWM spectrum. Thus, for some frequencies the stretched quadrature is amplified (bright fringes) while for others the squeezed quadrature is amplified (relatively dark fringes), as seen in Fig. 3a. The specific quadrature to be amplified can be controlled by the pump phase (see Methods for more details on the experiment). The broadband squeezing is evident already from the raw output spectrum, shown in Fig. 3a, where reduction of the parametric output below the vacuum noise level (the parametric output when the input is blocked) is observed across the entire 55 THz. To verify this, we varied the squeezing by varying the loss of the input FWM field before the measurement (second) pass through the PCF. As the loss is increased, the squeezing slowly vanishes, and even though the total power entering the fiber is diminished, the minimum fringes at the output of the measurement amplifier rise towards the vacuum input level, as shown in the inset of Fig. 3a.

The extraction of the quadrature information from the measured parametric output assumes knowledge of the parametric gain. The calibration of the parametric amplifier is simple,

performed by recording the output spectrum for a set of known inputs (Fig. 3b), when blocking various input fields (signal, idler, or pump). For example, the vacuum level of the parametric amplifier is observed when both the signal and the idler-input fields are blocked ($I_{zsi}$—zero signal idler). Also, the average number of photons at the input is given by the ratio of the measured output when the signal is blocked (idler only, $I_{zs}$—zero signal) to the vacuum input level $\langle N_i \rangle = \frac{I_{zs}}{I_{zsi}} - 1$. This calibration process is fully described in the Methods. After calibration, we obtain the parametric homodyne results of Fig. 3c, which show ~1.7 dB squeezing across the entire 55 THz bandwidth.

The observed squeezing in our experiment is far from ideal, primarily due to the fact that the pump is pulsed, which induces an undesirable time dependence of both the magnitude and phase of the parametric gain in the squeezing process, as well as in the parametric homodyne detection via self-phase and cross-phase modulation—SPM and XPM. Since our pump pulses are relatively long, their time dependence can be regarded as adiabatic, indicating that the instantaneous squeezing (source) and parametric amplification (measurement) are ideal, but the quadrature axis, squeezing level, and gain of the two amplifiers vary with time, not necessarily at the same rate. Thus, the measured spectrum, which represents a temporal average of the light intensity over the entire pulse, diminishes somewhat the expected squeezing (see illustration in the Methods).

Even with a pulsed pump, however, the various homodyne and calibration measurements are consistent and unequivocal for weak enough pump intensity (see Methods for further details on the pulse-averaging effects). With a pure CW pump, as is generally used in squeezing applications, this pulse-averaging limitation would not exist. Another limitation in our measurements is the need to re-couple the FWM back into the PCF, which introduces an inevitable loss of 30% and reduces the observed squeezing. This "known" loss can either be avoided completely in other experimental configurations or can be calibrated out to estimate the "bare" squeezing level of the measured light source (see Methods).

We verified the properties of the parametric homodyne detection in several ways. We measured the squeezed quadrature $\langle x^\dagger x \rangle$, and the uncertainty area, $\langle x^\dagger x \rangle \times \langle y^\dagger y \rangle$, of the squeezed state. Ideally, the generated squeezed light should be a minimum uncertainly state of $\langle x^\dagger x \rangle \times \langle y^\dagger y \rangle = 1$, independent of the generation gain; and the average intensity of the squeezed quadrature should exponentially decrease with the gain. The results are presented in the Methods, showing a clear reduction of the normalized squeezed quadrature intensity down to $\langle x^\dagger x \rangle \approx 0.68$ (32% below the vacuum level), and the uncertainty area remains nearly ideal at $\langle x^\dagger x \rangle \times \langle y^\dagger y \rangle < 1.3$, up to a pump power of

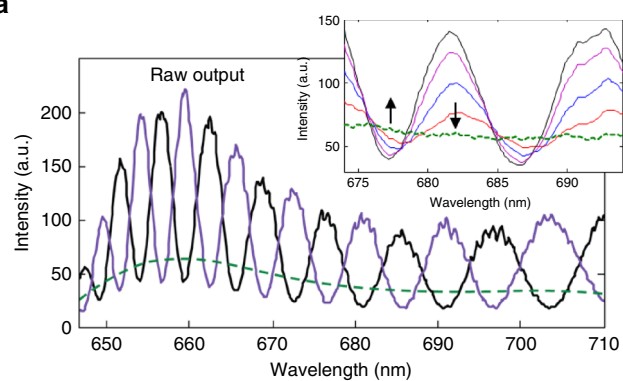

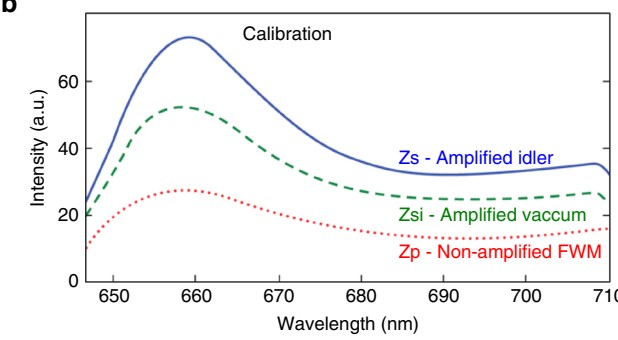

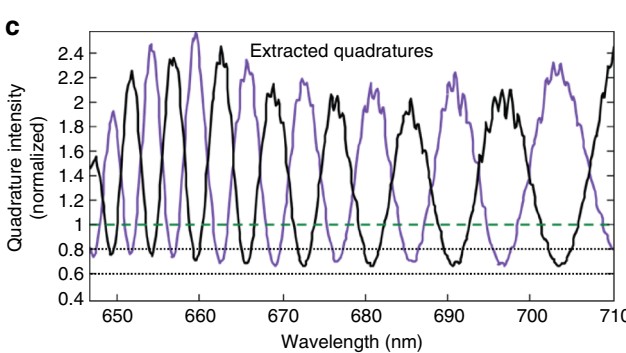

**Fig. 3** The procedure of parametric homodyne. Measurement of the quadratures includes three stages: **a** raw output measurement, **b** calibration, and **c** quadrature extraction. **a** Raw output measurements: In the most general case of arbitrary parametric gain, two measurements are needed to extract the quadrature information: (1) amplifying one quadrature (black); and (2) amplifying the orthogonal quadrature (purple). The specific quadrature to be amplified is defined by tuning the pump phase. The reduction of the raw output beneath the vacuum input level (dashed green) directly indicates squeezing. The inset shows the effect of loss at the input FWM light on the parametrically amplified output. As the loss is increased, the squeezing is reduced and the observed fringe minima rise towards the vacuum level (vertical arrows) even though the total input intensity is considerably decreased (a non-classical signature). **b** Calibration: To calibrate the parametric amplifier, the output response is measured for a set of three known inputs: (1) idler-input only (blocked signal, $I_{zs}$—solid blue), (2) vacuum input (blocked entire FWM—both signal and idler, $I_{zsi}$—dashed green), and (3) zero amplification (blocked pump, $I_{zp}$—dotted red). **c** Extracted quadratures (black and purple in accordance with **a**)—with the analysis detailed in the Methods, quadrature information is extracted. Quadrature squeezing is evident across the entire 55 THz spectrum down to $\langle x^\dagger x \rangle \approx 0.68$, 32% below the vacuum level (-1.7 dB)

60 mW. Further increase of the pump does not improve the measured squeezing due to pulse effects, and the minimum uncertainty property deteriorates. Based on the measured squeezing, the instantaneous squeezed quadrature at the peak of the pulse was estimated to be >3 dB (see Methods). Additional verification measurements of the broadband squeezing are presented and illustrated in the Methods.

**Two-mode quadratures**. The fundamental quadrature oscillation —a single-frequency component of a quadrature amplitude $x(\omega)$, $y(\omega)$, is a two-mode combination of frequencies $\omega_s = \Omega + \omega$ and $\omega_i = \Omega - \omega$—the signal and idler[25–27]. Using the field operators of the signal $a_s = a(\omega)$ and the idler $a_i = a(-\omega)$, the quantum operators of the quadratures $x(\omega)$, $y(\omega)$ are (see Methods for an intuitive reasoning)

$$\begin{cases} x(\omega) = a_s + a_i^\dagger \\ y(\omega) = i(a_s^\dagger - a_i) \end{cases}. \qquad (1)$$

This definition preserves the commutation relation $[x, y] = 2i$ and reduces in the monochromatic case to the single-mode quadratures $x = a + a^\dagger$, $y = i(a^\dagger - a)$.

The generalization of the standard quadratures to two-mode quadratures requires some attention. As opposed to the standard quadrature operators, which are hermitian and represent the time-independent real amplitude of the cosine (sine) oscillation, the two-mode quadrature operators of Eq. (1) are non-hermitian $x^\dagger(\omega) = x(-\omega) \neq x(\omega)$ and represent a time-dependent beat between the signal and idler modes with an envelope frequency $\omega$, carried by a cosine (sine) wave at frequency $\Omega$ (see Methods for some intuitive reasoning). The quadrature operators $x(\omega)$, $x^\dagger(\omega)$ represent the beat envelope, which has an amplitude and phase, in some similarity to the field operators $a$, $a^\dagger$ that represent the amplitude and phase of the carrier oscillation. Yet, the two-mode quadrature $x$ is an observable quantity (in contrast to the field operator $a$). Since $x$ commutes with its conjugate $[x(\omega), x^\dagger(\omega)] = 0$ (as opposed to $[a, a^\dagger] = 1$), it is possible to simultaneously measure both the real and imaginary part of the quadrature envelope, and thereby obtain complete information on both amplitude and phase of the single quadrature:

$$\begin{cases} \mathrm{Re}[x] = x + x^\dagger = X_s + X_i \\ \mathrm{Im}[x] = i(x - x^\dagger) = Y_s - Y_i, \end{cases} \qquad (2)$$

where $X_{s,i}$, $X_{s,i}$ are the standard single-mode quadratures of the signal and idler modes. Our experiment measured $x^\dagger(\omega)x(\omega)$.

Since the phase of the two-mode quadrature relates to commuting observables (as opposed to the carrier phase), it does not reflect a non-classical property of the quantum light field, but rather defines the classical temporal mode in which the field is measured. Specifically, the temporal mode of measurement is the two-frequency beat pattern of frequency $\omega$ (see Methods for an illustration), where the envelope phase defines the temporal offset of the beat. This offset, along with other mode parameters, such as polarization, spatial mode, carrier frequency, and so on define the mode of the LO. Of course, quantum entanglement is possible between the two envelope modes (cosine or sine) in direct equivalence to entanglement of a single photon (or photon pair, or cat state) between polarization modes, which is widely used for quantum information. However, this "quantumness" between modes is different and additional to the intra-mode quantum state, which is described by the quadratures $x$, $y$.

Due to the bandwidth limitation of standard homodyne measurement, the commonly used expression to interpret two-mode quadratures of optical frequency separation $\omega$ does not rely

on Eq. (1), but rather on Eq. (2). Two independent homodyne measurements of the signal and idler quadratures, $x_{s,i}$, $y_{s,i}$ need to be made relative to two correlated LOs at their respective frequencies $\omega_s$, $\omega_i$ so that the two output homodyne signals are within the electrical bandwidth. Thus, the standard procedure to measure just a single-frequency component of the two-mode quadrature $x(\omega)$ (and its squeezing) requires two separate homodyne measurements of the independent quadratures of both the signal and the idler using a pair of phase-correlated LOs[17,28]. For a broadband spectrum, standard two-mode homodyne requires a dense set of correlated pairs of LOs for each frequency component of the measurement. As we have shown, however, in our experiment above, a single LO is sufficient to simultaneously extract a specific quadrature across the entire optical bandwidth, just as a single pump laser can simultaneously generate the entire bandwidth of quadrature squeezed mode pairs.

**Quantum derivation of the parametric amplified output.** To model quantum mechanically the parametric homodyne process, we derive an expression for the parametric output intensity (photon-number) operator of the signal (or idler) mode, $N_s(g) = a_s^\dagger(g)a_s(g)$ ($g$ is the parametric gain) in terms of the input complex quadratures $x(\omega)$, $y(\omega)$. Mathematically, our method relies on the similarity between the quadrature operators of interest (Eq. (1)) $x(\omega) = a_s + a_i^\dagger$, $iy^\dagger = a_s - a_i^\dagger$ and the field operator at the output of a parametric amplifier:

$$a_s(g) = a_s \cosh(g) + e^{i\varphi} a_i^\dagger \sinh(g) \equiv Ca_s + Da_i^\dagger, \quad (3)$$

where the coefficients $C$ and $D$ are generally complex. Since field operators must fulfill $\left[a_s^\dagger(g), a_s(g)\right] = 1$, the two coefficients $C$ and $D$ must obey $|C|^2 - |D|^2 = 1$, which leads to the common description of $C = \cosh g$ and $D = e^{i\varphi} \sinh g$. However, the attributed phase of the parametric process $\varphi$, which is determined by the pump phase and the phase matching conditions in the nonlinear medium, can also be expressed explicitly, leaving the two coefficients $C$, $D$ real and positive (rather than complex), using $a_s(g,\theta) = \left(Ca_s e^{i\theta} + Da_i^\dagger e^{-i\theta}\right)e^{i\theta_0}$. Since the overall phase $\theta_0$ does not affect the photon-number calculations, we may discard it as $\theta_0 = 0$. In this expression we account for the phase of the pump as a rotation of the input quadrature axis—$a_{s,i} \to a_{s,i}e^{i\theta}$. Accordingly, the rotated complex quadrature operators (Eq. (1)) become $x_\theta(\omega) = a_s e^{i\theta} + a_i^\dagger e^{-i\theta}$ and $y_\theta(\omega) = i\left(a_s^\dagger e^{-i\theta} - a_i e^{i\theta}\right)$.

Parametric amplification directly amplifies one quadrature of the input and attenuates the other, as evident by expressing the field operators $a_s(g)$ at the output using the quadrature operators $x,y$ of the input:

$$a_s(g) = \frac{C+D}{2}x + i\frac{C-D}{2}y^\dagger = \frac{e^g}{2}x + \frac{e^{-g}}{2}iy^\dagger, \quad (4)$$

where the index $\theta$ was dropped for brevity. Finally, the parametric photon-number operator at the output is

$$\begin{aligned} N_s(g) = a_s^\dagger(g)a_s(g) &= \frac{N_s - N_i - 1}{2} \\ &+ \left(\frac{C+D}{2}\right)^2 x^\dagger x + \left(\frac{C-D}{2}\right)^2 y^\dagger y \\ &= \frac{N_s - N_i - 1}{2} + \frac{e^{2g}}{4}x^\dagger x + \frac{e^{-2g}}{4}y^\dagger y, \end{aligned} \quad (5)$$

where $N_{s,i} = a_{s,i}^\dagger a_{s,i}$ represent the input photon numbers (intensities) of the signal and idler. When access is available simultaneously to the intensities of both the signal and the idler, their sum of intensities provides the cleanest measurement of the

quadrature intensities

$$N_s(g) + N_i(g) = \frac{e^{2g}}{2}x^\dagger x + \frac{e^{-2g}}{2}y^\dagger y - 1. \quad (6)$$

Note that $N_s(g) - N_i(g) = N_s - N_i$ is a constant of the amplification, independent of the parametric gain.

With sufficient parametric gain, any given $x$ quadrature at the input can be amplified above the vacuum noise to a "classical level", even if it was originally squeezed, which allows complete freedom in measurement since vacuum fluctuations are no longer the limiting noise. If the measurement gain considerably exceeds the generation gain, such that $e^{2g}x^\dagger x \gg e^{-2g}y^\dagger y$, the amplified quadrature will dominate the intensity of the output light allowing to neglect the intensity of the attenuated orthogonal $y$ quadrature, and the measurement of the light intensity spectrum at the output will directly reflect (after calibration, see Methods) the single-shot value of the input quadrature intensity $x^\dagger(\omega)x(\omega)$, just like the standard measurement of the electrical spectrum at the output of standard homodyne.

Although the concept of parametric homodyne is conveniently understood in the limit of large gain, where the quadrature of interest dominates the output light field, parametric homodyne is equally effective with almost any finite gain. When the measurement gain is not large enough and the attenuated quadrature cannot be neglected, the two quadrature intensities can be easily extracted using a pair of measurements; setting the pump phase to amplify one quadrature ($\theta = 0$) and then to amplify the other ($\theta = \pi/2$), as illustrated in Fig. 3. Indeed, the output intensity in this case will not directly reflect the quadrature intensity, but it still provides equivalent information about the quadrature at any finite gain, since two light intensity measurements along orthogonal axes uniquely infer the two quadrature intensities at any finite gain, indicating that the information content of a measurement of the output intensity is the same as that of the quadrature intensity. An analytic derivation of this equivalence is provided in the Methods.

**Applicability to quantum tomography.** Quantum state tomography is a major application of homodyne measurement. It allows reconstruction of an arbitrary quantum state (or its density matrix or Wigner function) from a set of quadrature measurements along varying quadrature axes[7]. Unique reconstruction requires a complete measurement of the quadrature distribution function, which necessitates single-shot measurements of the instantaneous quadrature value, not just its average. Although both standard two-mode homodyne and parametric homodyne provide incomplete quadrature information in a single shot (in somewhat different ways), they still allow reconstruction of the quantum state under some assumptions. Hereon we review the different limitations of both methods and their implications to quantum tomography, leading to a conclusion that a combination of parametric homodyne followed by standard homodyne alleviates all the limitations and allows unambiguous reconstruction of arbitrary states.

Standard two-mode homodyne cannot provide a complete measurement of $x(\omega)$ in a single shot since standard homodyne is a destructive measurement. Specifically, observation of $\text{Re}[x(\omega)] = X_s + X_i$ requires a standard homodyne measurement of both frequency modes, which inevitably destroys the quantum state by photo-detection and prevents a consecutive measurement of $\text{Im}[x(\omega)] = Y_s - Y_i$. Splitting the state into two measurement channels is impossible since such a splitting will inevitably introduce additional vacuum noise. Thus, although $\text{Re}[x(\omega)]$ and $\text{Im}[x(\omega)]$ commute, standard two-mode homodyne can evaluate

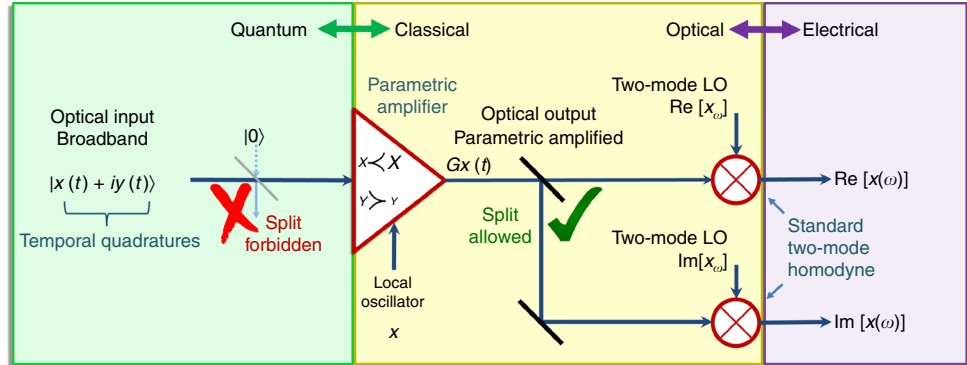

**Fig. 4** A Complete broadband homodyne scheme. Parametric homodyne allows to fully measure the two-mode quadrature amplitude; once the quadrature of interest $x(\omega)$ is sufficiently amplified above the vacuum level by the parametric amplifier, this quadrature becomes insensitive to additional vacuum noise, which allows splitting of the light into two standard homodyne channels in order to measure simultaneously both $\mathrm{Re}\,x(\omega)$ and $\mathrm{Im}\,x(\omega)$

only one of them in a single shot. In analogy to light polarization, standard homodyne acts as an absorptive polarizer that detects one polarization but absorbs the other, preventing complete analysis of the polarization state.

Our current realization of parametric homodyne suffers from a different ambiguity in a single shot (envelope phase). Since parametric homodyne measures only the instantaneous intensity of the quadrature $x^\dagger x$ (across a wide spectrum), but not its phase, only the probability distribution of the intensity $P(x^\dagger x)$ can be measured.

Let us analyze the ambiguity that is introduced to the reconstruction of a two-mode quantum state by the incomplete measurement, for both standard homodyne (only real part) and parametric homodyne (only intensity). For standard homodyne, the interpretation of a null result is ambiguous: a zero measurement can arise either from a true null of the measured quadrature or from a wrong selection of the envelope phase. Thus, standard homodyne can reconstruct a two-mode quantum state only if the envelope phase is fixed and known a priori. For two-mode squeezed vacuum, however, which is the major two-mode quantum state that is experimentally accessible, the envelope phase is random, indicating that standard homodyne can provide only the average fluctuations $\langle x^\dagger x \rangle = \langle (X_s + X_i)^2 \rangle + \langle (Y_s - Y_i)^2 \rangle$, but not the single shot value of the quadrature (or its intensity).

For parametric homodyne, where the quadrature intensity is measured, null (or any intensity) is unambiguously interpreted for any envelope phase, but the sign of the measured quadrature is ambiguous. Thus, complete reconstruction is possible (for any envelope phase) only if the symmetry of the quadratures is known, which is relevant to a large set of important quantum states. For example, photon-number states or squeezed states[6] that are known to be symmetric can be reconstructed, and indeed the non-classicality of a single photon state is directly manifested by the fact that the probability to measure a null intensity vanishes $P\left(x_\theta^\dagger x_\theta = 0\right) = 0$ for any quadrature axis $\theta$, which inevitably indicates negativity of the Wigner function at zero field. Yet, a two-mode coherent state $|\pm\alpha\rangle$ and cat states like $|\alpha\rangle \pm |-\alpha\rangle$[8] can be differentiated only if the symmetry of the state is assumed a priori. For broadband squeezed vacuum, where the envelope phase is inherently random, this measurement is ideal.

Clearly, the two methods complete each other in their capabilities, indicating that a combination of parametric homodyne with interferometric detection is the perfect solution to a complete measurement, as illustrated in Fig. 4. Specifically, parametric gain is a non-demolition process (contrary to

standard homodyne) that provides a light output and allows extraction of the complete quadrature information in a single shot, including the phase. Thus, if the measured quadrature is amplified sufficiently above the vacuum, this quadrature becomes insensitive to loss, even for moderate gain values. The parametric output light can thus be split to two homodyne channels that will measure both $\mathrm{Re}[x(\omega)]$ and $\mathrm{Im}[x(\omega)]$ simultaneously (see Fig. 4). The splitting does not hamper the measurement (contrary to standard homodyne) since the added vacuum affects primarily the attenuated quadrature, which is not measured.

In the literature, the possibility to add a parametric amplifier before electronic detection was analyzed in several different contexts: already the seminal paper of Caves from 1981 that introduced squeezed vacuum to the unused port of an interferometer for sub-shot noise interferometric measurement, suggested to include a parametric amplifier in the detection arm to overcome the quantum inefficiency of photo-detectors[29], Leonhardt and Paul[30] later suggested a similar use of parametric amplification for quantum tomography that is insensitive to loss, Ralph[31] suggested it for teleportation and Davis et al.[32] for the analysis of atomic spin-squeezing. Most recently, this concept was experimentally implemented for atomic spin measurements in[33] enabling phase detection down to 20 dB below the standard quantum limit with inefficient detectors.

**Comparison to standard homodyne**. It is illuminating to examine on equal footing standard homodyne measurement and the parametric homodyne method. After all, the balanced detection in standard homodyne produces a down-converted RF field at the difference-frequency of the two optical inputs (LO and signal), similar to optical down-conversion, which is the core of parametric amplification. In that view, the well-known homodyne gain of balanced detection (proportional to the LO field) produces an amplified electronic version of the input quantum quadrature, directly analogous to the parametric gain (proportional to the pump amplitude), which optically amplifies a single input quadrature. Thus, both the standard homodyne gain and the optical parametric gain serve the same homodyne purpose—to amplify the quantum input of interest (the optical quadrature) to a classically detectable output level[34], which is sufficiently above the measurement noise (the electronic noise for standard homodyne or the optical vacuum noise for parametric homodyne). Consequently, standard and parametric homodyne are two faces of the same concept.

The difference between the two schemes is both technical and conceptual. On the technical level, the gain of standard homodyne is generally very large, allowing to a priori neglect

any effect of the unmeasured quadrature on the electrical output, whereas the optical parametric gain may not be sufficient to justify such an a priori assumption and may require more careful analysis of the output with finite gain, as we described earlier. On the conceptual level, parametric homodyne provides an optical output, as opposed to standard homodyne that destroys the optical fields. Since the optical parametric output can be sufficiently "classical" (amplified above the vacuum level), it is far less sensitive to additional vacuum noise from optical loss or detector inefficiency. Consequently, parametric homodyne does not only preserve the optical bandwidth across the quantum-classical transition (see Fig. 1), but can also allow complete reconstruction of the two-mode quadrature in a single shot, as was explained in the previous sub-section. Hence, adding a layer of optical parametric gain before the electronic photo-detection, be it intensity detection or homodyne provides an important freedom to quantum measurement beyond the ability to preserve the optical bandwidth.

**Beyond the pure two-mode field.** Last, let us briefly consider broadband time-dependent states of light beyond the single-frequency two-mode state. Any classical wave packet with spectral envelope $f(\omega) = |f(\omega)|e^{i\varphi(\omega)}$ around the carrier frequency $\Omega$ (normalized to $\int d\omega |f(\omega)|^2 = 1$) can be regarded as an electromagnetic mode with associated quantum field operators

$$\begin{cases} a_f(t) = \int d\omega f(\omega)a(\omega)e^{-i\omega t} \\ a_f^\dagger(t) = \int d\omega f^\star(\omega)a^\dagger(\omega)e^{i\omega t}, \end{cases} \quad (7)$$

and associated temporal quadratures

$$\begin{cases} x_f(t) = \int d\omega e^{-i\omega t}\left[f(\omega)a(\omega) + f^\star(-\omega)a^\dagger(-\omega)\right] \\ y_f(t) = i\int d\omega e^{-i\omega t}\left[f^\star(-\omega)a^\dagger(-\omega) - f(\omega)a(\omega)\right], \end{cases} \quad (8)$$

which is just the Fourier transform of Eq. (1) (see also Eq. (11) in the Methods).

We can express the temporal quadrature $x_f(t)$ in terms of the two-mode quadratures $x(\omega)$, $y(\omega)$ as

$$\begin{cases} x_f(t) = \int d\omega e^{-i\omega t}\left[\frac{f(\omega)+f^\star(-\omega)}{2}x(\omega) + i\frac{f(\omega)-f^\star(-\omega)}{2}y^\dagger(\omega)\right] \\ y_f(t) = \int d\omega e^{-i\omega t}\left[\frac{f(\omega)+f^\star(-\omega)}{2}y^\dagger(\omega) - i\frac{f(\omega)-f^\star(-\omega)}{2}x(\omega)\right], \end{cases} \quad (9)$$

where the symmetric and antisymmetric parts of the wave packet $\frac{f(\omega)+f^\star(-\omega)}{2}, \frac{f(\omega)-f^\star(-\omega)}{2}$ are the Fourier transforms of Re$f(t)$, Im$f(t)$ the real and imaginary parts of the field envelope in time.

Equation (9) can be simplified considerably when the spectrum of the wave packet is symmetric $|f(\omega)| = |f(-\omega)|$, which is the major situation to employ a quadrature representation to begin with. The temporal quadrature $x_f(t)$ is then simply a superposition of many two-mode components $x_\theta(\omega)$ with a spectrally varying axis $\theta(\omega)$ and envelope phase $\delta(\omega)$

$$\begin{cases} x_f(t) = \int d\omega e^{-i\omega t}|f(\omega)|e^{i\delta(\omega)}x_{\theta(\omega)}(\omega) \\ = \int d\omega e^{-i\omega t}|f(\omega)|e^{i\delta(\omega)}\left[x(\omega)\cos\theta(\omega) + y^\dagger(\omega)\sin\theta(\omega)\right] \\ y_f(t) = \int d\omega e^{-i\omega t}|f(\omega)|e^{i\delta(\omega)}y_{\theta(\omega)}^\dagger(\omega) \\ = \int d\omega e^{-i\omega t}|f(\omega)|e^{i\delta(\omega)}\left[y^\dagger(\omega)\cos\theta(\omega) - x(\omega)\sin\theta(\omega)\right]. \end{cases} \quad (10)$$

The quadrature axis of each two-mode component is dictated by its carrier phase $\theta(\omega) = \frac{\varphi(\omega)+\varphi(-\omega)}{2}$—the symmetric part of the spectral phase of the wave packet $\varphi(\omega)$; and the two-mode

envelope phase $\delta(\omega) = \frac{\varphi(\omega)-\varphi(-\omega)}{2}$ relates to the antisymmetric part of $\varphi(\omega)$. Thus, for a transform limited mode, where $\varphi(\omega) = 0$, both the envelope phase and the quadrature axis are constant across the spectrum $\delta(\omega) = 0$, $\theta(\omega) = 0$. An antisymmetric phase variation, $(\varphi(\omega) = -\varphi(-\omega))$, will affect only the envelope phase, but keep the quadrature axis constant $\theta(\omega) = 0$, as is the case for down-converted light. A purely symmetric phase $\varphi(\omega) = \varphi(-\omega)$, as due to material dispersion, will affect only the quadrature axis, but keep the envelope constant $\delta(\omega) = 0$.

Therefore, measurement of an arbitrary generalized quadrature of broadband light requires measurement (or knowledge) of two spectral degrees of freedom—the quadrature axis $\theta(\omega)$ and the envelope phase $\delta(\omega)$. Parametric homodyne with intensity measurement provides complete information of the quadrature axis $\theta(\omega)$ (by measuring the output spectrum for varying pump phase), but is insensitive to $\delta(\omega)$. It therefore allows measurement if $\delta(\omega)$ is either unimportant (down-conversion) or known a priori (transform limit or well-defined pulse), which is relevant to all current sources of broadband quantum light in spite of the limitations. The combination of parametric gain followed by standard homodyne allows complete arbitrary measurement, as explained above.

## Discussion

It is interesting to note that the effect of two parametric amplifiers in series was deeply explored previously in the context of quantum interference[35]. In such a series configuration, interference occurs between two possibilities for generating bi-photons, either in the first amplifier or in the second, depending on the pump phase. The interference contrast can reach unity when the parametric gain of the two amplifiers is identical (assuming no loss), which testifies to the quantum nature of the light in both the single-photon regime[21] and at high power[20,35]. Here, however, we consider the second amplifier as a measurement device, independent of the source of light to be measured. This light source can be, but is certainly not limited to be, a squeezing parametric amplifier. Clearly, any other source of quantum light is relevant when homodyne measurement is of interest, such as single photons, Fock states, NOON states, Schrödinger cat states, and so on.

A different optical measurement of quantum light was recently reported in ref. [36], where vacuum fluctuations of THz radiation were observed in time. There too an optical nonlinearity (of several THz bandwidth) was utilized for a direct measurement, where the large bandwidth of the nonlinearity was key to enable time sampling of the vacuum fluctuations, well within a single optical cycle of the measured THz mode.

To conclude, we presented an approach to optical homodyne measurement with practically unlimited bandwidth, which adds a layer of optical parametric amplification before the photo-detection, and enables simultaneous quadrature measurement across the entire spectrum with a single LO. This measurement removes major limitations of optical homodyne and opens a wide window for efficient utilization of the bandwidth resource for parallel quantum information processing. An interesting expansion of this concept would be where the pump itself includes more than one mode, for measurement of "hyper" entanglement between different frequency pairs of the frequency comb with a multi-mode pump[13,37].

## Methods

**Two-mode quadratures: time and frequency representation.** The direct mathematical relation of the time varying field to broadband quadrature amplitudes is simple and illuminating in both time and frequency, and yet, it is rarely used outside the context of near monochromatic light. For a classical time-dependent field $E(t) = a(t)\exp i\Omega t +$ c.c., the two quadratures in time are the real

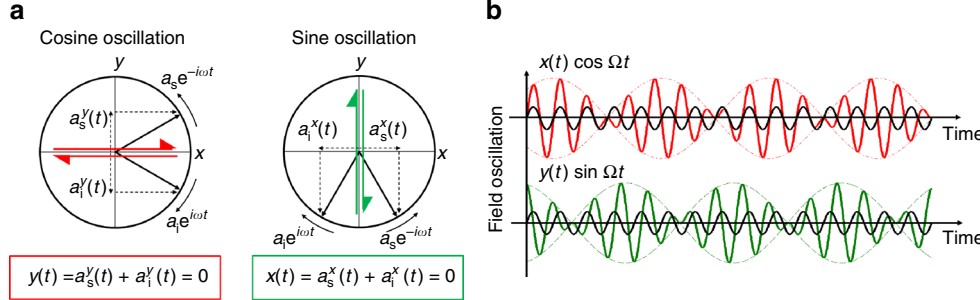

**Fig. 5** Time-domain illustration of a pure two-mode quadrature oscillation with a single-frequency component. **a** Quadrature maps of the signal and idler field amplitudes $a_s, a_i$ in a rotating frame at the carrier frequency $\Omega = \frac{\omega_s + \omega_i}{2}$. Due to the frequency difference from the carrier, the amplitudes of the signal and idler rotate around the quadrature map in opposite directions at rates $+\omega = \omega_s - \Omega$ and $-\omega = \omega_i - \Omega$. If the two amplitudes are equal in magnitude ($|a_s| = |a_i|$), they constantly cancel out along one quadrature axis, and add up along the orthogonal axis to form an oscillating beat—the pure quadrature oscillation; for the cosine oscillation $a_s^y(t) + a_i^y(t) = 0$, and for the sine oscillation $a_s^x(t) + a_i^x(t) = 0$. **b** Time-domain illustration of pure two-mode quadrature oscillations, showing the two-mode beat envelope at frequency $\omega$ modulating the carrier frequency oscillation. The quadrature axis is defined by the phase $\frac{\varphi_s + \varphi_i}{2}$ of the carrier of the two-mode oscillation relative to a reference LO (black) –0, $\pi$ for the $x(\omega)$ quadrature (cosine) and $\pm \frac{\pi}{2}$ for $y(\omega)$. The phase of the quadrature amplitude $\frac{\varphi_s - \varphi_i}{2}$ reflects the temporal offset of the beat envelope with respect to a time reference

and imaginary parts of the field amplitude $a(t)$

$$\begin{cases} x(t) = a(t) + a^*(t) = 2\,\mathrm{Re}[a(t)], \\ y(t) = i[a^*(t) - a(t)] = 2\,\mathrm{Im}[a(t)]. \end{cases} \quad (11)$$

In frequency, therefore, the quadrature amplitudes $x(\omega)$, $y(\omega)$, represent the symmetric and antisymmetric parts of the field spectral amplitude $a(\omega)$

$$\begin{cases} x(\omega) = a(\omega) + a^*(-\omega), \\ y(\omega) = i[a^*(\omega) - a(-\omega)], \end{cases} \quad (12)$$

where $\omega$ is the offset from the carrier frequency $\Omega$, possibly of optical separation, and $a(\omega) = \frac{1}{\sqrt{2\pi}} \int a(t) e^{-i\omega t} dt$. Strictly speaking, Eqs. (1) and (12) define the quadratures of the nonlinear dipole within the medium, not of the emitted light field. Specifically, they do not include the frequency dependence of the optical field operator $E(\omega) \sim a(\omega)\sqrt{\omega}$, which is different for the signal and idler modes. Yet, to avoid cumbersome nomenclature we simply refer to these as the "two-mode quadratures," since they correctly represent the quantum correlation and squeezing of a two-mode field.

Figure 5 illustrates the temporal field of a single two-mode component of a pure quadrature oscillation, which represents a beat pattern: slow sinusoidal envelope of frequency $\omega$ over a fast carrier wave at frequency $\Omega$ (cosine or sine). The temporal two-mode field can be written in terms of the two-mode quadratures as

$$\begin{aligned} E_{\Omega,\omega}(t) =\ & \left[ a_s e^{-i(\Omega + \omega)t} + a_i e^{-i(\Omega - \omega)t} \right] + \text{c.c.} \\ =\ & \left[ x(\omega) e^{-i\omega t} + x^\dagger(\omega) e^{i\omega t} \right] \cos \Omega t \\ & + \left[ y(\omega) e^{-i\omega t} + y^\dagger(\omega) e^{i\omega t} \right] \sin \Omega t \end{aligned} \quad (13)$$

where the terms in the square brackets represent the quadrature envelopes.

**Calibration of the parametric amplifier.** The parametric amplifier must be calibrated to allow extraction of the quadrature information from the measured output intensities (using Eqs. (5) or (6)). In our experiment, the spectral width that could be detected by the spectrometer was limited to ~100 nm, which prevented measurement of both the signal and the idler simultaneously. In addition, the detection efficiency for the idler frequencies was considerably reduced compared to the signal. Thus, we measured mostly the signal band and relied on Eq. (5) to obtain the quadratures.

Generally, five calibration parameters are required—the gain coefficients $|C|$ and $|D|$ (without the phases that define the quadrature axis), the average photon numbers of the two input modes $\overline{N}_s, \overline{N}_i$ (to evaluate the offset term $\frac{N_s - N_i - 1}{2}$ in Eq. (5)), and the overall detector response per single photon $n_0^2$. Thus, five independent measurements are required. In many applications, however, the number of independent measurements may be reduced, since the offset term may be treated as just $-\frac{1}{2}$, when the photon-number difference is zero, which is generally zero for squeezed light with symmetric loss; and the parametric amplifier may be assumed ideal ($|C|^2 = |D|^2 + 1$) if the gain is not very high.

Using Eq. (3), the measurement output (proportional to the FWM intensity) is

$$\begin{aligned} I_s =\ & n_0^2 \big[ |C|^2 \overline{N}_s + |D|^2 (\overline{N}_i + 1) \\ & + C^* D \langle a_s^\dagger a_i \rangle + C D^* \langle a_s a_i^\dagger \rangle \big]. \end{aligned} \quad (14)$$

For calibration we use measurements that are independent of phase-coherent terms ($\langle a_s^\dagger a_i \rangle = \langle a_s a_i^\dagger \rangle = 0$ or $D = 0$), allowing us to write $I_s = n_0^2 \big[ |C|^2 \overline{N}_s + |D|^2 (\overline{N}_i + 1) \big]$. We first measure the output intensity in two scenarios: (1) $I_{zsi}$, blocking the signal and idler (vacuum input) and (2) $I_{zs}$, blocking the signal (only idler input). The names of the calibration quantities are chosen in analogy to the engineering formalism for evaluating linear systems by measuring their response in various cases, termed: zero input response and zero state response. We use similar indices for the various parametric responses: zero signal (ZS), zero idler (ZI), zero signal and idler (ZSI), and zero pump (ZP). These measurements provide (with the aid of Eq. (14)) $I_{zsi} = n_0^2 |D|^2$ and $I_{zs} = n_0^2 |D|^2 (\overline{N}_i + 1)$, indicating that the ratio between these two measurements yields the idler average photon-number $\overline{N}_i = \frac{I_{zs}}{I_{zsi}} - 1$. Note that these two measurements act as a simple method for acquiring the input number of photons independent of the parametric gain. The signal photon-number $\overline{N}_s = \frac{I_{zi}}{I_{zsi}} - 1$ can be acquired by measuring the output idler intensities in the same way (or be assumed equal to $\overline{N}_i$, if appropriate).

Next, we use the knowledge of the input photon numbers for calibrating the overall detector response $n_0^2$. We measure: (3) $I_{zp}$, blocking the pump (zero amplification, $|C| = 1$, $|D| = 0$, letting the signal and idler through). Again, from Eq. (14) we find $n_0^2 = \frac{I_{zp}}{N_s}$.

Once the detector response is obtained, we can obtain the parametric gain coefficients $|C|, |D|$ with the $I_{zsi}$ measurement, since $|D|^2 = \frac{I_{zsi}}{n_0^2}$, and $|C|^2 = \frac{|D|^2}{N_s} \left( \frac{I_{zsi}}{I_{zsi}} + 1 \right)$ (or may be assumed ideal $|C|^2 = |D|^2 + 1$, if appropriate). Note that the calibration is needed only once, as long as the parametric measurement gain is constant, and the average photon-number difference at the input $\overline{N}_i - \overline{N}_s$ does not change (typically for squeezed input, this difference is simply zero).

**Extraction of the average quadratures.** The two quadratures cannot be measured simultaneously, but their average intensities can both be extracted from two measurements of the parametric output intensity, amplifying one quadrature first ($I_x$) and then the other ($I_y$), according to

$$\begin{cases} \langle x^\dagger x \rangle = \frac{1}{r^2 - q^2} \left[ r(I_x/n_0^2 - p) - q(I_y/n_0^2 - p) \right] \\ \langle y^\dagger y \rangle = \frac{1}{r^2 - q^2} \left[ r(I_y/n_0^2 - p) - q(I_x/n_0^2 - p) \right], \end{cases} \quad (15)$$

where $n_0^2$ is the detector response per single photon and the coefficients $p$, $q$, and $r$ are:

$$\begin{cases} p = \frac{1}{2}(\overline{N}_s - \overline{N}_i - 1) \\ q = \frac{1}{4}(|C| + |D|)^2 \\ r = \frac{1}{4}(|C| - |D|)^2. \end{cases} \quad (16)$$

**Parametric homodyne with finite gain.** To consider more formally the equivalence of parametric amplification to extraction of quadrature information at any finite gain, let us examine the relation between the field operators at the output of the amplifier and the quadratures of the input (Eq. (3)) $a_s(\theta, g) = a_s e^{-i\theta} \cosh(g) + a_i^\dagger e^{i\theta} \sinh(g) = x_\theta e^g + iy_\theta^\dagger e^{-g}$. As mentioned, the field operator converges in the limit of large gain to an amplified single quadrature operator $a_s(\theta, g) \to e^g x_\theta$, but this convergence can never be exact since the commutation relation of field

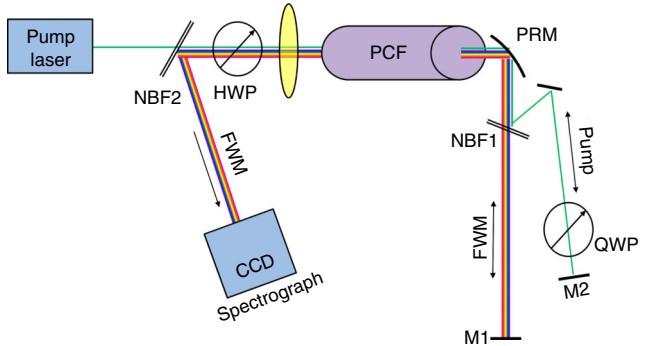

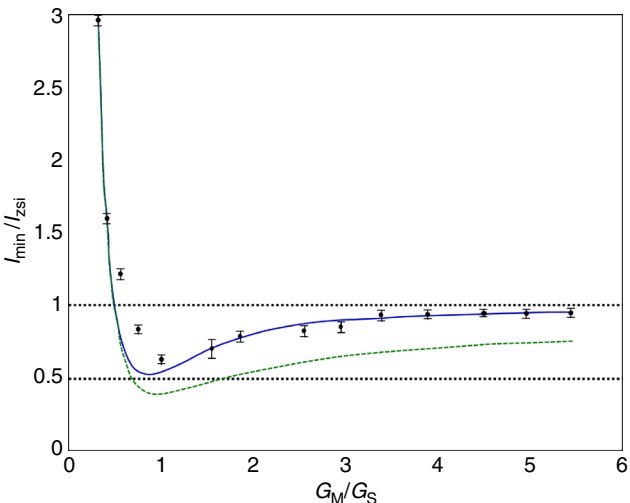

**Fig. 6** Details of the experimental setup. The experiment consisted of two parts: (1) generation of broadband squeezed light (propagating from left to right) and then (2) homodyne measurement of the generated squeezing (propagating back from right to left). Broadband two-mode squeezed light is generated via spontaneous four-wave mixing (FWM) in a photonic crystal fiber (PCF) pumped by 12 ps laser pulses (786 nm) (collimated at the output with a parabolic mirror PRM). After generation, the pump and the FWM are separated into two paths by a narrowband filter (NBF1) allowing the pump phase and polarization to be tuned independently. Both pump and FWM are reflected back by folding mirrors (M1, M2) for a second pass through the PCF, now acting as a measuring device. The specific FWM quadrature to be amplified is selected by tuning the pump phase, and the amplification gain is controlled by manipulation of the pump polarization using the half-wave and quarter-wave plate (HWP, QWP) before the fiber and after it. After the second (measurement) pass through the PCF, the FWM field is separated from the pump by a narrowband filter (NBF2) and directed towards a CCD spectrograph, where the amplified spectrum is measured

operators $[a, a^\dagger] = 1$ is inherently different than that of quadrature operators $[x, x^\dagger] = 0$. To illuminate the smooth transition from a field operator to a quadrature, let us express the field operator for any finite parametric gain in the form of a generalized quadrature operator along an axis of a complex angle $\vartheta = \theta + i\gamma$,

$$a_s(g) = M(\tilde{x}\cos\vartheta + \tilde{y}^\dagger\sin\vartheta) = M\tilde{x}_\vartheta, \qquad (17)$$

where the imaginary part of the quadrature axis and the normalization factor $M$ relate to the gain $g$ by $\tanh\gamma = e^{-2g}, M^2 = 2/\sinh 2\gamma$.

Thus, the single-shot measurement of the output light intensity with any parametric gain reflects the intensity of the "generalized" quadrature at this gain value, and not the standard (real) quadrature. The commutation relation of these generalized quadratures is

$$\left[\tilde{x}_\vartheta, \tilde{x}_\vartheta^\dagger\right] = 1/M^2 \approx e^{-2g}, \qquad (18)$$

where the approximation is valid already for moderate gain of $g \geq 1$. Consequently, the commutator of the measured generalized quadratures, converges very quickly to that of the real quadratures.

**Details of the experimental setup.** In our experiment (Figs. 2 and 6), we generate an ultra-broadband two-mode squeezed vacuum via collinear FWM in a PCF, which is pumped by narrowband 12 ps pulses at 786 nm with up to 100 mW average power. The broad bandwidth is obtained by closely matching the pump wavelength to the zero dispersion of the fiber at 784 nm[21], resulting in a signal and idler bandwidth of ~55 THz each, with ~90 THz mean frequency separation between the mode centers (700 nm—signal center, and 900 nm—idler center). After generation, the pump is separated from the FWM field into a different optical path by a narrowband filter (NBF1—Semrock NF03-808E-25), allowing independent control of the relative pump phase. The pump phase is actively locked to the phase of the FWM using an electro-optic modulator and a fast feedback loop. Both the FWM and pump fields are reflected back (mirrors M1, M2) towards the PCF for a second pass, which then acts as the homodyne measurement. The final parametric amplified spectrum (after the second "homodyne" pass) is filtered from the pump (NBF2—Semrock NF03-785E-25) and measured with a cooled CCD spectrograph (SpectraPro 2300i).

In order to partially compensate for the temporal pulse effects due to SPM of the pulsed pump, we used the original pump pulse from the first pass through the PCF also for the second pass. This guaranteed that the pump and the FWM accumulated nearly the same phase modulation (either SPM for the pump or XPM for the FWM light). Polarization manipulations were used to tune the effective

**Fig. 7** Dependence of the squeezed fringe on the parametric gain. The squeezed quadrature can be directly obtained from the relative output of the minimum fringe when the measurement gain ($G_M$) is large enough (relative to the squeezing gain—$G_S$)—the ratio between the output with input ($I_{min}$) and the output with vacuum (blocked) input ($I_{zsi}$). For increased measurement gain (but constant input squeezing), the relative parametric output should therefore converge to a constant level, directly indicating the absolute quadrature intensity. To observe this, we varied the pump intensity in the second (measurement) pass up to 5.5 times the intensity used for generating the broadband squeezing in the first (generation) pass. This convergence of the relative output (dots) $\left(\frac{G_M}{G_S} > 3\right)$ approached a constant level of ~5% below the vacuum level. When the measurement gain is reduced, the relative output decreases due to a quantum interference effect, reaching maximum visibility when the squeezing generation gain is equal to the measurement gain $\left(\frac{G_M}{G_S} = 1\right)$. In this regime, the general measurement becomes indirect (although the squeezing effect is still directly evident), and a pair of measurements (amplifying one quadrature and then the other) is needed for extracting the quadrature information. Below the level of identical gain, the observed output strongly rises over the vacuum level, obscuring the direct evidence of squeezing; however, the quadrature information can still be extracted (though with reduced accuracy) using the same pair of measurements. The solid (blue) curve indicates a numerical simulation of the relative output, assuming the measured pump pulse energy and FWM loss, and an estimated nonlinear coefficient, fiber coupling efficiency, and hyperbolic-secant pulse shape. For comparison, we included the simulated result for the relative output at the peak of the pulse (dashed green)

parametric gain in the second (measurement) pass independently of the squeezing strength in the first pass: since the phase matching conditions in the PCF are polarization dependent, the observed FWM spectrum is generated only by one polarization of the pump (this fact was extensively verified).

Thus, rotating the pump polarization before the first pass with a half-wave plate we could transfer part of the pump power through the fiber without affecting the FWM. This power could later be used in the second pass by rotating its polarization back to the PCF axis with a quarter-wave plate in the pump beam path. This extra pump power accumulated almost the same SPM as the FWM, but without affecting the squeezing generation.

The various calibration measurements were performed by manipulating the FWM light between the passes either by physically blocking the FWM beam (vacuum input) or pump beam (zero amplification) or with a high-efficiency optical long-pass filter (idler-input only) (Semrock FF776-Dio1). The two orthogonal homodyne measurements (amplifying the squeezed quadrature or the stretched quadrature) were acquired by tuning the offset of the active feedback loop that locked the pump phase.

**Effects of the pulsed pump.** In our experiment, the pump for both generation of the squeezed light and for the parametric homodyne measurement (second pass) is a pulsed laser of ≈12 ps duration. Since the bandwidth of the generated FWM (55

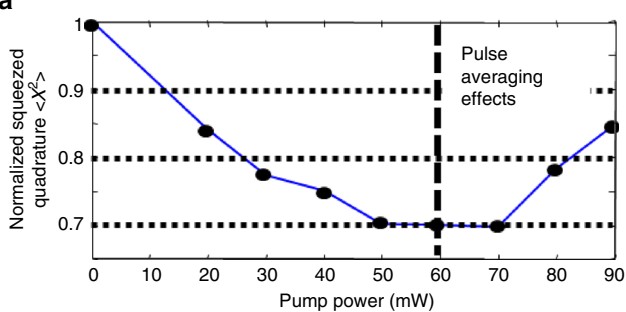

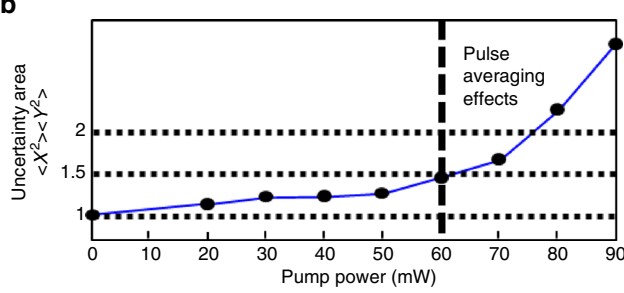

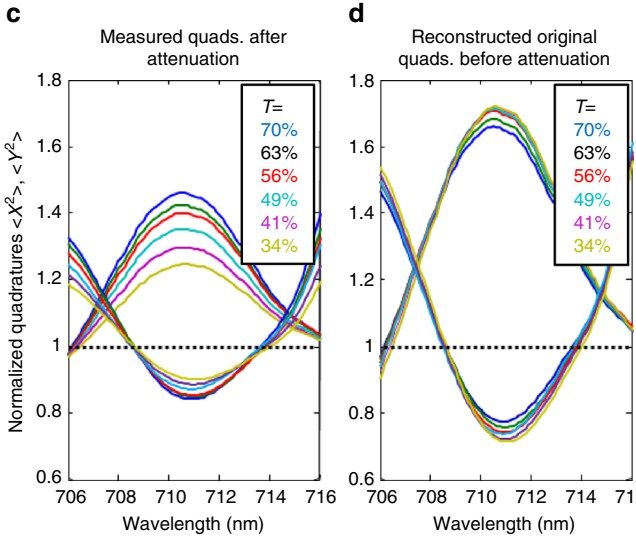

**Fig. 8** Expanded homodyne results. **a** Measured squeezed quadrature as a function of squeezing strength (696 nm)—(solid line for guidance only). As the squeezing strength in the first pass is increased, the measured squeezed quadrature decreases, down to $\langle x^\dagger x \rangle \sim 0.68$ at a pump power of ~ 60 mW. Further increase of the pump degrades the observed squeezing due to temporal effects of the pulsed pump. **b** Minimum uncertainty conservation (696 nm)—(solid line for guidance only). Ideal squeezed vacuum is a minimum uncertainty state of $\langle x^\dagger x \rangle \langle y^\dagger y \rangle = 1$, independently of the squeezing strength. Up to a pump power of 60 mW, the uncertainty area is indeed nearly conserved ($\langle x^\dagger x \rangle \langle y^\dagger y \rangle < 1.3$). Beyond this limit the pulse-averaging effect washes out the minimum uncertainty property. **c, d** The effect of loss on the squeezed state. **c** We apply a series of loss values (30–66%) to a given squeezed state and observe the influence on the squeezed/stretched quadratures. **d** The reconstructed "bare" squeezed/stretched quadratures that calibrated out the loss from all the curves of **c**, demonstrating collapse of all the curves to nearly the same value, as expected

THz) is much larger than the pump bandwidth (<0.1 THz), we could account for the main affect of the pulse shape as an adiabatic variation of the parametric gain and phase modulation (SPM, XPM) along the temporal profile of the pump pulse. Thus, the adiabatic variation can be discretized in time, referring to time instances within a single pulse as separate parametric events of varying gain and phase.

However, since the integration time of the photo-detectors in the CCD spectrograph is much longer (~10 ms), the measured homodyne data is averaged over the entire shape of many pulses.

The effect of the pulse on the parametric gain alone changes the generated squeezing and the measurement gain with time, measuring weak squeezing with weak parametric gain at the edges of the pulse, and strong squeezing with strong parametric gain at the peak. The phase modulation (SPM, XPM) of the FWM process has a more severe effect, since it modulates in time the quadrature axis to be amplified. As a result, due to the pump pulse shape, the amplified quadrature axis of the FWM field rotates with time. Luckily, when the pump itself experiences nearly the SPM it can still act as a near-perfect LO (phase regarding) for measuring the FWM, even after passage through the fiber. The small residual difference between the pump SPM and the FWM XPM causes the amplified FWM quadrature to rotate with time, mixing different quadrature axes together in the same measurement, smearing out some of the squeezing.

Ideally, we would like to extract the maximum squeezing that occurs at the peak of the pulse from the time-averaged measurements. To estimate this peak squeezing, we numerically simulated the entire FWM generation and parametric amplification along the pump pulse with 50 fs temporal resolution (corresponding to the coherence time of the FWM). The simulation incorporated the measured pump pulse energy, the measured loss and fiber coupling efficiencies, and an assumed hyperbolic-secant temporal shape of the pump pulse (12 ps). Using the simulation, we could calculate both the average and the peak outputs of the process, allowing us to estimate the squeezing at the peak of the pulse from the measured averaged homodyne output. Figure 7 demonstrates the relation between the peak homodyne output and the average homodyne output, as the parametric measurement gain is varied. As long as the generation pump power does not exceed a specific limit (~60 mW in our experiment), the pulse-averaging only affects the absolute measured squeezing values (which can be roughly estimated) but not the expected trends of the experiment (increasing the loss, the squeezing power, or the parametric power).

**Expanded results**. To verify the properties of the parametric homodyne, we measured the quadrature squeezing $\langle x^\dagger x \rangle$, and the uncertainty area, $\langle x^\dagger x \rangle \times \langle y^\dagger y \rangle$ of the squeezed state as described in the main text.

Another important verification of our squeezing measurement is to observe the effect of loss on the measured quadrature squeezing and stretching. We measured the quadrature intensities after applying a set of known attenuations (30–66% loss), and reconstructed the "bare" quadratures before loss, which indeed collapsed to the same value, as shown in Fig. 8c, d. The effect of loss on the quadrature intensity can be regarded as propagation through a beam splitter with one open port. The relations between the operators of the two inputs ($a_1$, $a_2$) and two outputs ($b_3$, $b_4$) of the beam splitter can be defined as $b_3 = ta_1 + ra_2$ and $b_4 = ta_1 - ra_2$, where $t$ and $r$ are the transmission and reflection (loss) amplitudes. In these terms, the quadrature operator at output port 3 is: $x_3 = tx_1 + rx_2$, and the expectation value of the quadrature intensity is

$$\langle x_3^2 \rangle = |t|^2 \langle x_1^2 \rangle + |r|^2 \langle x_2^2 \rangle + 2rt \langle x_1 \rangle \langle x_2 \rangle. \quad (19)$$

Assuming a vacuum state at the open input port 2, the final expression becomes:

$$\langle x_3^2 \rangle = |t|^2 \langle x_1^2 \rangle + |r|^2. \quad (20)$$

Hence, the "bare" quadratures, before the loss, can be reconstructed using

$$\langle x^\dagger x \rangle_{\text{bare}} = \left( \langle x^\dagger x \rangle_{\text{measured}} - |r|^2 \right) / |t|^2. \quad (21)$$

As a complementary evaluation, we studied the parametric measurement-amplifier output as a function of its own gain, while maintaining the squeezing generation gain constant. For this, we gradually increased the pump power in the second pass up to 5.5 times the pump power that generated the squeezing in the first pass. When the parametric gain is strong enough, the output intensity relative to the vacuum level (without input) is directly proportional to the input quadrature. Hence, we expect the relative output to stabilize as the parametric gain is increased, and indeed the observed reduction below the vacuum level stabilized at 5%. Figure 7 shows the measured results and addresses the pulse effects on this measurement.

**Data availability**. All relevant data are available from the authors.

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

## Acknowledgements

This research was funded by the "Bikura" (FIRST) program of the Israel science foundation (ISF grant #44/14).

## Author contributions

Y.S., A.P., and M.R. designed the experiment, Y.S. and A.P. developed the theory, Y.S., Y.M., and R.Z.V. performed the experiment, L.B. participated in the analysis of results and continuous discussions. All authors contributed to finalizing the manuscript.

## Additional information

**Competing interests:** The authors declare no competing financial interests.

