## [Peer Review File · Nature Communications]

Reviewers' comments:

Reviewer #1 (Remarks to the Author):

After reading the new version of the manuscript, I am convinced that the achievement reported in it is of sufficient caliber to warrant publication in Nature Communications. Of particular significance is the argument that the proposed method permits single-shot measurements of a quadrature, albeit a different one compared to conventional homodyne detection.

The presentation quality has also significantly improved. My comments from the previous review have been addressed satisfactorily. However, I still found it difficult to read the manuscript, primarily because the authors use different language compared to what I am used to. My expertise is in quantum optical homodyne tomography, and the notions of sine and cosine quadratures, carrier and envelope phase, which the authors use a lot in their work, are not too common in this community. I think the paper may become more accessible to if the authors found a way to reduce the use of this language, or at least provide better explanation thereof. An example to that effect (among many others) is Eq. (3) and the sentence immediately preceding it. The equation would become clearer if it started with an expression in terms of the operators $a(+\omega)$ and also included an expression for $y_\omega(t)$. In the subsequent sentence, one should relate a_s and a_i to $a(+\omega)$.

An additional issue is that the main body of the manuscript does not explain the principle of the method deeply enough. It only states (in line 4 on page 4) the quantity being measured, but defers the explanation to the Methods. I believe that the body of the paper should contain more rigorous discussion so the reader can understand the authors' idea without studying the Methods, especially since the authors already made a great effort to define the formalism on page 2. For example, moving Eq. (7) to the body of the paper would be helpful.

I am also a bit concerned about the phrase on page 4 that the measurement "does not provide the phase of the quadrature envelope". This may mislead the reader into thinking that the method is not phase sensitive. This is obviously not the case: otherwise the authors would not be able to observe squeezing. In this context, it would be good if the authors could tell whether the set of quantities they measure can be used to calculate the time-dependent instantaneous quadratures $x(t)$ and $p(t)$. If this is not possible, how do the authors envision performing tomography of e.g. Fock states in the time domain (page 4 line 74)?

That being said, I reiterate my opinion that this journal is an appropriate venue for the manuscript. While I recommend that the authors do more work on the presentation, I do not want this to become an obstacle to publication.

Reviewer #3 (Remarks to the Author):

The submitted paper introduces a novel technique that allows homodyne measurements over a very wide frequency range, far exceeding the current state-of-the-art. The results are quite impressive, showing e.g. a measurement of squeezing over a wavelength range of tens of nanometers.

From what I can tell from the correspondence that was sent with the submission, there have been significant changes to the article already trying to clarify its message. One of the fundamental difficulties seems to be the distinction between two-mode homodyne (using two phase-stabilised local oscillators) and single-mode homodyne (using just a single local oscillator), where the authors claim that their method should be compared to the former, and that this is the current standard in e.g. cv quantum information. I would disagree with that, in that the usual key-distribution schemes (e.g. the schemes by Grosshans, Nature 421, 238; Gehring, NatComm 8795 as referenced in the correspondence) use just a single-mode local oscillator and rely on the (limited) detection bandwidth around its frequency. Figure 1 compares the authors' new scheme to the conventional "standard" homodyne, but also this picture shows single-mode homodyne instead of the two-mode homodyne that the authors want to be compared with. Compared to single-mode homodyne, the technique is however inferior, as was pointed out by the previous referees, and I believe that their points are still valid. While the authors point at extensions and new schemes around their technique which might recreate some of the features of single-mode homodyne, these are not (yet) experimentally proven and thus cannot actually replace the unique features of single-mode homodyne in a wide variety of applications.

Now, I enjoy the idea and results and would like to see this work published in some form. However, it still feels like the article does not clearly and succinctly capture the idea that they propose. The article is very technical and probably became even more technical with the explanations that were added as a result of reviewer comments. The introduction is still very long, and much of the supplemental material is in fact required reading to be able to understand the conclusions that the authors draw. While the authors provide an interesting outlook on QKD with their setup, as part of the correspondence, the outlook is not part of the paper at hand and can do little to increase its usefulness to a wide audience. In its current form, I am afraid the article is not a good fit for Nature Communications.

Referee #1:

Comment: *After reading the new version of the manuscript, I am convinced that the achievement reported in it is of sufficient caliber to warrant publication in Nature Communications. Of particular significance is the argument that the proposed method permits single-shot measurements of a quadrature, albeit a different one compared to conventional homodyne detection.*

The presentation quality has also significantly improved. My comments from the previous review have been addressed satisfactorily. However, I still found it difficult to read the manuscript, primarily because the authors use different language compared to what I am used to. My expertise is in quantum optical homodyne tomography, and the notions of sine and cosine quadratures, carrier and envelope phase, which the authors use a lot in their work, are not too common in this community. I think the paper may become more accessible to if the authors found a way to reduce the use of this language, or at least provide better explanation thereof. An example to that effect (among many others) is Eq. (3) and the sentence immediately preceding it. The equation would become clearer if it started with an expression in terms of the operators $a(+\omega)$ and also included an expression for $y_\omega(t)$. In the subsequent sentence, one should relate a_s and a_i to $a(+\omega)$.

Our response: We thank the referee for the positive judgement of the manuscript. Indeed, we found that for a sufficient discussion of broadband phenomena, we were required to employ some of the jargon of ultrafast optics, which is far less common in the quantum optics community. However, we fully accept the referee's comment, since eventually, our major target audience is the quantum optics community.

Resulting modifications:

1. We added figure 4 in the revised manuscript to illustrate two-mode oscillations in time, and to clarify the notion of carrier phase and envelope phase.
2. We made an intensive effort to define all operators in a unified language as early as possible, and to clarify the 'ultrafast' jargon, wherever it appears.

Comment: *An additional issue is that the main body of the manuscript does not explain the principle of the method deeply enough. It only states (in line 4 on page 4) the quantity being measured, but defers the explanation to the Methods. I believe that the body of the paper should contain more rigorous discussion so the reader can understand the authors' idea without studying the Methods, especially since the authors already made a great effort to define the formalism on page 2. For example, moving Eq. (7) to the body of the paper would be helpful.*

Our response: We fully accept the comment, which was raised also by referee #3. We rearranged the body of the article and expanded it to include all the major theoretical derivations that previously appeared in the 'methods'.

Resulting modifications: Our results are now organized in the following subsections:

1. Experiment
Moved to start the results. Slightly modified to facilitate understanding before the theoretical background.
2. Theoretical Foundation
Includes the previous description of quadratures in time / frequency and definition of quadrature operators.
3. Two-Mode Quadratures
Includes the previous comparison to degenerate quadratures. Added figure 4 to illustrate the new degree of freedom – the envelope phase.
4. Quantum Derivation of the Parametric Amplified Output Intensity
Moved from 'methods'
5. Parametric Homodyne with Finite Gain
Moved from 'methods'
6. Applicability to Quantum Tomography
Moved from 'methods'. Expanded and added new figure 5 to explain and illustrate the advantages of combining parametric homodyne with standard homodyne.
7. Comparison to Standard Homodyne
No change.
8. Beyond the Single-Frequency Two-Mode Field
New subsection to illustrate tomographic reconstruction of arbitrary broadband quantum states of light.

Comment: *I am also a bit concerned about the phrase on page 4 that the measurement “does not provide the phase of the quadrature envelope”. This may mislead the reader into thinking that the method is not phase sensitive. This is obviously not the case: otherwise the authors would not be able to observe squeezing. In this context, it would be good if the authors could tell whether the set of quantities they measure can be used to calculate the time-dependent instantaneous quadratures $x(t)$ and $p(t)$. If this is not possible, how do the authors envision performing tomography of e.g. Fock states in the time domain (page 4 line 74)?*

Our response: We accept, and made a great effort to clarify.

Resulting modifications: We added figure 4 to illustrate both the carrier phase and the envelope phase, and the difference between them. The new subsection 'Beyond the Single-Frequency Two-Mode Field' discusses in some detail the reconstruction of generalized temporal quadratures of arbitrary temporal fields from measurements of parametric homodyne.

Comment: That being said, I reiterate my opinion that this journal is an appropriate venue for the manuscript. While I recommend that the authors do more work on the presentation, I do not want this to become an obstacle to publication.

Our response: We thank the referee for the positive judgement.

Referee #3:

Comment: *The submitted paper introduces a novel technique that allows homodyne measurements over a very wide frequency range, far exceeding the current state-of-the-art. The results are quite impressive, showing e.g. a measurement of squeezing over a wavelength range of tens of nanometers.*

From what I can tell from the correspondence that was sent with the submission, there have been significant changes to the article already trying to clarify its message. One of the fundamental difficulties seems to be the distinction between two-mode homodyne (using two phase-stabilized local oscillators) and single-mode homodyne (using just a single local oscillator), where the authors claim that their method should be compared to the former, and that this is the current standard in e.g. cv quantum information. I would disagree with that, in that the usual key-distribution schemes (e.g. the schemes by Grosshans, Nature 421, 238; Gehringer, NatComm 8795 as referenced in the correspondence) use just a single-mode local oscillator and rely on the (limited) detection bandwidth around its frequency. Figure 1 compares the authors' new scheme to the conventional "standard" homodyne, but also this picture shows single-mode homodyne instead of the two-mode homodyne that the authors want to be compared with. Compared to single-mode homodyne, the technique is however inferior, as was pointed out by the previous referees, and I believe that their points are still valid. While the authors point at extensions and new schemes around their technique which might recreate some of the features of single-mode homodyne, these are not (yet) experimentally proven and thus cannot actually replace the unique features of single-mode homodyne in a wide variety of applications.

Our response: As reviewer #3 claims, the common homodyne method, based on the electrical nonlinearity of square-law photo-detectors, uses a single frequency LO. As so, and as a result of the bandwidth limitation of the photo-detectors, all common homodyne applications are bandwidth limited to near monochromatic signals, within the MHz to GHz range (so called single-mode states). Within this range standard single-mode homodyne enables the slowly varying quadrature information to be followed temporally, maintaining the time dependence of the information.

Beyond single-mode states, are two-mode states (pairs of narrowband modes that are widely separated in frequency), and broadband states (a continuum of mode pairs). The only existing method to address two-mode states, is by applying two single frequency homodyne measurements in parallel, one for each frequency mode, using two LO frequencies – i.e. standard two-mode homodyne. For broadband states of optical bandwidth, standard homodyne methods offer no solution, thus, such states could not be measured at all, until now.

Hence, almost all homodyne applications are severely limited in bandwidth due to the narrowband response of the electronic nonlinearity of the photo-detectors. *We emphasize through examples in the manuscript the great potential that broadband homodyne can offer. It is this limitation that we aim to lift. Thus, we do not claim to offer a better solution to single-mode homodyne; Rather, we aim to offer a solution at all, even if partial, for two-mode and broadband homodyne measurement, for which no solution is possible with standard methods.* Hence, for two-mode measurement, parametric homodyne capability must be compared to standard two-mode homodyne; whereas in the broadband regime, no relevant comparison exists.

It is also very important to consider parametric homodyne as an intermediate layer of quadrature-specific pre-amplification prior to a standard homodyne measurement, even for single-mode detection. As was recognized several times in the past (see references 29-34 in the manuscript), such a quadrature pre-amplifier allows detection with inefficient detectors, and simplifies the measurement considerably. Recently, this concept was used to detect phenomenal levels of spin-squeezing (>20dB) in cold atoms (Hosten, O. *et al* "Quantum phase magnification" *Science* **352**, 1552-1555 (2016)).

With regards to figure 1, the main objective of the figure is to explain the optical operation of parametric homodyne. Since parametric homodyne uses a single LO to amplify the broadband oscillation quadrature as a whole, it performs in the optical regime (and with optical bandwidth) the same operation that single-mode homodyne performs in the electrical regime. We find it therefore accurate to illustrate the measurement concepts on equal footing. The differences between parametric homodyne and single-mode homodyne arise from the electrical detection stage, which is not illustrated in figure 1 (but is addressed in the text, and now also in the new figures 4 and 5).

Resulting modifications: Figure 5 was added to illustrate possibilities of electrical detection within parametric homodyne. The new subsection '*Beyond the Single-Frequency Two-Mode Field*' discusses in some detail the reconstruction of generalized temporal quadratures of arbitrary temporal fields from measurements of parametric homodyne. A new paragraph was added (page 9) to the subsection 'applicability to quantum tomography', which details previous considerations from the literature of pre-amplification of a quadrature of interest before detection in various contexts. This paragraph now reads:

“In the literature, the possibility to add a parametric amplifier before electronic detection was analyzed in several different contexts: Already the seminal paper of Caves from 1981 that introduced squeezed vacuum to the unused port of an interferometer for sub-shot noise interferometric measurement, suggested to include a parametric amplifier in the detection arm to overcome the quantum inefficiency of photo-detectors [30], Leonhardt later suggested a similar use of parametric amplification for quantum tomography that is insensitive to loss [31], Ralph suggested it for teleportation [32] and Davis et al for analysis of atomic spin-squeezing [33]. Most recently, this concept was experimentally implemented for atomic spin measurements in [34] enabling phase detection down to 20dB below the standard quantum limit with inefficient detectors.”

Comment: Now, I enjoy the idea and results and would like to see this work published in some form. However, it still feels like the article does not clearly and succinctly capture the idea that they propose. The article is very technical and probably became even more technical with the explanations that were added as a result of reviewer comments. The introduction is still very long, and much of the supplemental material is in fact required reading to be able to understand the conclusions that the authors draw. While the authors provide an interesting outlook on QKD with their setup, as part of the correspondence, the outlook is not part of the paper at hand and can do little to increase its usefulness to a wide audience. In its current form, I am afraid the article is not a good fit for Nature Communications.

Our response: We accept the comment. The body of the article was rearranged to shorten the introduction considerably, and to move all the major theory components from the 'methods' to the main text, as was detailed above in the response to referee #1.

REVIEWERS' COMMENTS:

Reviewer #3 (Remarks to the Author):

Review "Lifting the Bandwidth Limit of Optical Homodyne Measurement"

I have received the above paper after significant changes have been made to its contents based on earlier comments from another reviewer and myself. Previously, many explanations and connections were not clearly laid out and so left the reader a bit lost, in terms of what to make of the results and where the demonstrated concept can be applied. This is now much better, unfortunately at the cost of a much longer manuscript, which I assume will be significantly over the word count limit for an article. I do believe, however, that shortening the paper is possible without losing on content. The current wealth of information has, indeed, allowed me to understand the paper and its implications much better. I therefore would like to make the following comments and recommendations.

As I see it, the new and striking science in the presented work is a direct measurement of quadrature modulations at and below the shot-noise limit over a very wide bandwidth. The underlying principle of parametric amplification to push noise way above shot-noise, such that detection loss can be neglected, is not new. This is also acknowledged by the authors in the newly added paragraph, going back to an initial suggestion by Caves in 1981. In addition to the references given by the authors, I think it's also widely used in microwave experiments with micro-mechanical oscillators. Usually, however, the amplification stage is then followed by some sort of standard homodyne detection. It is the direct detection of the amplified and mixed sidebands that allows the impressively wide bandwidth of the present experiment.

Now, in my opinion much of the paper tries to explain known science in a sometimes cumbersome way and could be vastly reduced.

As a start, the notation adapted in Eq (3) is commonly known as "two-photon formalism" in the field of interferometry and quantum optomechanics (starting with the back-to-back publications of Caves and Schumaker, one of which is cited here as [27]). It is a standard way to describe (quantum) modulation sidebands. Thus, the content of "Theoretical Foundation" and "Two-Mode Quadratures" could be reduced to just a quick reminder of two-photon formalism and a few references. Fig. 4 is not needed.

I'm not exactly sure why $\text{Re}[x]$ and $\text{Im}[x]$ are introduced, as I would have thought that the usual amplitude and phase quadrature operators x and y are sufficient for the description that follows.

I would argue that "single-mode homodyne" as referred to by the paper is actually a two-mode process when using the operators of the two-photon formalism, which is necessary when speaking about modulation sidebands. I.e., "single-mode homodyne" senses the two modes $a(\Omega \pm \omega)$. Then, I don't see why "two-mode homodyne"/"standard homodyne" is necessary for the paper (although it should of course still be referenced). While it allows to span a wider bandwidth than a single-LO homodyne, I don't think it provides additional insight, especially since homodyne detection with two local oscillators does not seem all that "standard" to me and therefore it might confuse readers more than it helps.

I still think that Fig. 1 is not very helpful, mostly because all the waveforms make it look extremely busy.

The section on "Quantum Derivation of the Parametric Amplified Output Intensity" is the theoretical gist of the article, and it is well explained. Together with the experimental results, it does however raise the question about $N_i(g, \theta)$, i.e., was the "idler" part of the spectrum measured as well? From the given wavelength range it seems that this was not the case, but due to the symmetry of Eq (6) for signal and idler, there should be valuable information in the idler part as well. Is there anything to be gained in e.g. electronically combining both parts? Also, maybe some intuitive understanding could be added into where exactly the difference between standard homodyne (measuring $g \cdot x$) and parametric homodyne (measuring $g \cdot (x^\dagger x)$) comes from, something that I think Fig. 1 was meant to provide, but at least for me it doesn't.

The section "Parametric Homodyne with Finite Gain" seems like a good candidate to go into a Methods section (again?).

"Applicability to Quantum Tomography" makes a few helpful statements, and the references to amplification before detection are a useful addition, as stated above. However, the statement about the random envelope phase in two-mode squeezed vacuum states seems to contradict standard experimental practise at least in the continuous-wave regime, where tomography of such states is routinely performed using phase-stable reference beams (requiring that the source produces a steady stream of identical copies of the quantum state over the whole measurement time)? Of course, as is pointed out in this section and the next, the measurement output of parametric homodyne is still useful in many cases.

Care should be taken about the wording at the end of "Comparison to Standard Homodyne" - "adding a layer of optical parametric gain before the electronic photo-detection [...] provides a fundamental new freedom to quantum measurements"; as stated in the previous section, adding a parametric amplifier before the detection is not new, and its use in optimising quantum measurements has been studied extensively in e.g. SU(1,1) interferometers [Manceau et al 2017 New J. Phys. 19 013014].

All in all, I feel that still quite a bit of editing effort should go into the paper at hand, but given the solid science and strong results I would then recommend it for publication in Nature Communications.

I waive my right to anonymity.

Sebastian Steinlechner

Manuscript NCOMMS-17-12589A – Response to referee reports

Referee (Dr. Steinlechner):

Comment: *I have received the above paper after significant changes have been made to its contents based on earlier comments from another reviewer and myself. Previously, many explanations and connections were not clearly laid out and so left the reader a bit lost, in terms of what to make of the results and where the demonstrated concept can be applied. This is now much better, unfortunately at the cost of a much longer manuscript, which I assume will be significantly over the word count limit for an article. I do believe, however, that shortening the paper is possible without losing on content. The current wealth of information has, indeed, allowed me to understand the paper and its implications much better. I therefore would like to make the following comments and recommendations.*

Our response: We thank the referee for the positive judgement of the manuscript. We agree that the paper was long, but as the referee pointed out, most of the added information was important in order to understand the paper and its implications. We therefore shortened the main text along the lines requested by the referee (as detailed hereon), but maintained most of the information, shifting some of it to the Methods, for the readers.

Comment: *As I see it, the new and striking science in the presented work is a direct measurement of quadrature modulations at and below the shot-noise limit over a very wide bandwidth. The underlying principle of parametric amplification to push noise way above shot-noise, such that detection loss can be neglected, is not new. This is also acknowledged by the authors in the newly added paragraph, going back to an initial suggestion by Caves in 1981. In addition to the references given by the authors, I think it's also widely used in microwave experiments with micro-mechanical oscillators. Usually, however, the amplification stage is then followed by some sort of standard homodyne detection. It is the direct detection of the amplified and mixed sidebands that allows the impressively wide bandwidth of the present experiment.*

Our response: We agree with the referee. Indeed, the broadband direct detection after parametric amplification is the major message of our manuscript.

Comment: *Now, in my opinion much of the paper tries to explain known science in a sometimes cumbersome way and could be vastly reduced.*

As a start, the notation adapted in Eq (3) is commonly known as "two-photon formalism" in the field of interferometry and quantum optomechanics (starting with the back-to-back publications of Caves and Schumaker, one of which is cited here as [27]). It is a standard way to describe (quantum) modulation sidebands. Thus, the content of "Theoretical Foundation" and "Two-Mode Quadratures" could be reduced to just a quick reminder of two-photon formalism and a few references. Fig. 4 is not needed.

Our response: Indeed, the “two-photon formalism” is well known and standard for quantum modulation sidebands, but was so far applied only within the framework of the electronic bandwidth. For the optical regime it was never used, because it could not be directly linked to the experimental capabilities. We therefore find that a short review will still be helpful for readers, which now appears only in the Methods.

Resulting modifications: The subsections "Theoretical Foundation" and "Two-Mode Quadratures" were unified to a single subsection and shortened considerably. The intuitive derivation of the two-mode quadrature operators, was moved to the Methods along with figure 4 (now fig. 5) under the subsection “Two-Mode Quadratures: Time and Frequency Representation”

Comment: *I'm not exactly sure why $Re[x]$ and $Im[x]$ are introduced, as I would have thought that the usual amplitude and phase quadrature operators x and y are sufficient for the description that follows.*

Our response: $Re[x] = x + x^\dagger$, $Im[x] = i(x - x^\dagger)$ are introduced because x itself is non Hermitian and cannot be measured directly. $Re[x]$, $Im[x]$ are the only observables that can be measured experimentally, and luckily, they commute $[Re[x], Im[x]] = 0$ (contrary to the quadratures themselves $[x, y] = 2i$), which allows to measure both $Re[x]$, $Im[x]$ simultaneously and obtain indirectly the complete information of a single quadrature in a single shot, as we explained. This is therefore an important and unique feature of two-mode quadratures, which does not exist for standard single-mode quadratures, and we find it necessary to present it.

Resulting modifications: We find that figure 5 is important to convey the message of two-mode oscillations and their envelope phase. We therefore maintained it in the Methods.

Comment: *I would argue that "single-mode homodyne" as referred to by the paper is actually a two-mode process when using the operators of the two-photon formalism, which is necessary when speaking about modulation sidebands. I.e., "single-mode homodyne" senses the two modes $a(\Omega \pm \omega)$. Then, I don't see why "two-mode homodyne"/"standard homodyne" is necessary for the paper (although it should of course still be referenced). While it allows to span a wider bandwidth than a single-LO homodyne, I don't think it provides additional insight, especially since homodyne detection with two local oscillators does not seem all that "standard" to me and therefore it might confuse readers more than it helps.*

Our response: It is correct of course that the "two-photon formalism" applies equally to standard single mode homodyne as it does to parametric homodyne. The main difference is that in single-mode homodyne, the frequency difference between the modes is low enough to allow direct observation of the slow beat in time, whereas for broadband parametric homodyne, the only possible observation is in frequency. The concept of envelope amplitude and phase (or real and imaginary parts) becomes critical here, as opposed to standard homodyne. Furthermore, till now, the only available method to measure two-mode homodyne with broad separation was with two local oscillators. This method may be less common than single-mode homodyne, but for those who attempt to use the optical bandwidth resource it is the gold standard, such as Olivier Pfister who aims to utilize the quantum frequency comb for quantum information processing.

Comment: *I still think that Fig. 1 is not very helpful, mostly because all the waveforms make it look extremely busy.*

Our response: We accept, and tried to reduce the number of waveforms to the necessary minimum – Input and output fields, each of them decomposed into two quadrature components.

Resulting modifications: Figure 1 was revised. The spectral decomposition was modified to be less busy, and the background color was toned down to brighten the image.

Comment: *The section on "Quantum Derivation of the Parametric Amplified Output Intensity" is the theoretical gist of the article, and it is well explained. Together with the experimental results, it does however raise the question about $N_i(g, \theta)$, i.e., was the "idler" part of the spectrum measured as well? From the given wavelength range it seems that this was not the case, but due to the symmetry of Eq (6) for signal and idler, there should be valuable information in the idler part as well. Is there anything to be gained in e.g. electronically combining both parts? Also, maybe some intuitive understanding could be added into where exactly the difference between standard homodyne (measuring g^*x) and parametric homodyne (measuring $g^*(x^\dagger)$) comes from, something that I think Fig. 1 was meant to provide, but at least for me it doesn't.*

Our response: We thank the referee for this important question. Indeed by summing the signal and the idler intensities, it is possible to null the input-dependent offset terms of equation 5 and obtain a cleaner measurement of the quadrature intensities. In our experiment however, the detectable spectral width of the spectrometer was limited to ~100nm, which prevented measurement of both the signal and the idler simultaneously. In addition, the detection efficiency for the idler

frequencies was considerably reduced compared to the signal. Thus, we measured mostly the signal band and relied on equation 5 to obtain the quadratures with calibration.

Resulting modifications:

We added equation 6 and reworded the surrounding paragraph to spell out the direct relation of the intensity-sum to the amplified quadrature. The subsection in the Methods on calibration of the parametric amplifier was also modified to lay out the experimental considerations.

Comment: *The section "Parametric Homodyne with Finite Gain" seems like a good candidate to go into a Methods section (again?).*

Our response: Accepted (again).

Resulting modifications: The subsection was moved to the Methods.

Comment: *"Applicability to Quantum Tomography" makes a few helpful statements, and the references to amplification before detection are a useful addition, as stated above. However, the statement about the random envelope phase in two-mode squeezed vacuum states seems to contradict standard experimental practice at least in the continuous-wave regime, where tomography of such states is routinely performed using phase-stable reference beams (requiring that the source produces a steady stream of identical copies of the quantum state over the whole measurement time)? Of course, as is pointed out in this section and the next, the measurement output of parametric homodyne is still useful in many cases.*

Our response: The attempt to draw analogy from CW experiments is misleading. As mentioned, a two-mode squeezed vacuum obtains two degrees of freedom in the optical phase – the carrier phase and the envelope phase (as opposed to CW squeezed vacuum, which relates only to the carrier phase). In terms of the carrier phase, two-mode squeezing does generate routinely identical copies of the quantum state, whereas for the envelope phase it is random. This does not hamper the parametric homodyne measurement, since parametric amplification is blind to envelope phase and does not measure it at all.

Resulting modifications: We find that figure 5 is important to convey the message of two-mode oscillations, their envelope phase, and the difference from standard CW quadratures. We therefore maintained it in the Methods.

Comment: *Care should be taken about the wording at the end of "Comparison to Standard Homodyne" - "adding a layer of optical parametric gain before the electronic photo-detection [...] provides a fundamental new freedom to quantum measurements"; as stated in the previous section, adding a parametric amplifier before the detection is not new, and its use in optimizing quantum measurements has been studied extensively in e.g. SU(1,1) interferometers [Manceau et al 2017 New J. Phys. 19 013014].*

Our response: Accepted.

Resulting modifications: We reworded the statement, which now reads "...adding a layer of optical parametric gain before the electronic photo-detection, be it intensity detection or homodyne provides an important freedom to quantum measurement beyond the ability to preserve the optical bandwidth".

Comment: *All in all, I feel that still quite a bit of editing effort should go into the paper at hand, but given the solid science and strong results I would then recommend it for publication in Nature Communications.*

Our response: We thank the referee for the positive judgement of the manuscript. We are optimistic that the current revision